# Changes in RNA Splicing: A New Paradigm of Transcriptional Responses to Probiotic Action in the Mammalian Brain

**DOI:** 10.3390/microorganisms13010165

**Published:** 2025-01-14

**Authors:** Xiaojie Yue, Lei Zhu, Zhigang Zhang

**Affiliations:** State Key Laboratory for Conservation and Utilization of Bio-Resources in Yunnan, School of Life Sciences, Yunnan University, Kunming 650091, China; 18738301103@163.com (X.Y.); zhulei_evan@126.com (L.Z.)

**Keywords:** probiotics, gut–brain axis, RNA splicing, psychiatric disorders

## Abstract

Elucidating the gene regulatory mechanisms underlying the gut–brain axis is critical for uncovering novel gut–brain interaction pathways and developing therapeutic strategies for gut bacteria-associated neurological disorders. Most studies have primarily investigated how gut bacteria modulate host epigenetics and gene expression; their impact on host alternative splicing, particularly in the brain, remains largely unexplored. Here, we investigated the effects of the gut-associated probiotic Lacidofil^®^ on alternative splicing across 10 regions of the rat brain using published RNA-sequencing data. The Lacidofil^®^ altogether altered 2941 differential splicing events, predominantly, skipped exon (SE) and mutually exclusive exon (MXE) events. Protein–protein interactions and a KEGG analysis of differentially spliced genes (DSGs) revealed consistent enrichment in the spliceosome and vesicle transport complexes, as well as in pathways related to neurodegenerative diseases, synaptic function and plasticity, and substance addiction across brain regions. Using the PsyGeNET platform, we found that DSGs from the locus coeruleus (LConly), medial preoptic area (mPOA), and ventral dentate gyrus (venDG) were enriched in depression-associated or schizophrenia-associated genes. Notably, we highlight the *App* gene, where Lacidofil^®^ precisely regulated the splicing of two exons causally involved in amyloid β protein-based neurodegenerative diseases. Although the splicing factors exhibited both splicing plasticity and expression plasticity in response to Lacidofil^®^, the overlap between DSGs and differentially expressed genes (DEGs) in most brain regions was rather low. Our study provides novel mechanistic insight into how gut probiotics might influence brain function through the modulation of RNA splicing.

## 1. Introduction

The gut–brain axis, which gradually formed through prolonged coevolution between animals and their symbiotic microbes, enables bidirectional interplay between gut bacteria and the host brain [1,2,3] through multiple routes including the autonomic nervous system, the enteric nervous system, the endocrine system, immune processes, and metabolic fluxes [4]. The homeostasis of gut bacterial composition and metabolic activity is an important basis for host brain activity, emotional states, cognitive processes, and behavioral patterns [5,6,7,8].

Nowadays, demographic aging and the prevalence of unhealthy lifestyles have prompted a dramatic worldwide increase in the number of patients with neurological diseases, causing neurological diseases to emerge as the primary contributor to global ill-health [9,10,11]. However, effective diagnosis of and therapy for these disorders are still facing huge challenges, due to the extreme complexity of the nervous system [12,13]. With the rise in clinical and epidemiological evidence, gastrointestinal diseases, including inflammatory bowel disease, irritable bowel syndrome, and eosinophilic esophagitis, are frequently observed to persistently co-occur with common mental illnesses [14,15], highlighting the potential role of the gut–brain axis in such disease comorbidity.

A wide range of model animals (e.g., *C. elegans*, *Drosophila*, zebrafish, and mice) and human pathological samples have facilitated scientists to deeply explore the molecular mechanisms underpinning the gut–brain axis [16,17,18,19,20]. The gut microbiota has been shown to influence host brain transcription through multiple mechanisms. The metabolites derived from gut bacteria affect the brain’s epigenetic landscape, including DNA methylation and diverse histone modifications [21], through regulating the activity of epigenetic modification enzymes such as DNA methyltransferases and histone deacetylases [22,23,24]. These epigenetic alterations could further influence chromatin restructuring and the recruitment of transcription factors, thereby inhibiting or activating gene transcription [25]. Moreover, microRNAs partially mediate the regulation of host brain gene expression by gut bacteria. For example, significant alterations in microRNA expression profiles were observed in the hippocampus of germ-free mice, and these molecular changes were partially reversed following subsequent gut bacteria colonization [26].

RNA splicing, qualitatively regulating protein structure and functionality [27], is particularly abundant in brain tissue. It empowers the brain to construct a highly intricate neuronal network with extremely limited genes [28]. Recently, the abnormalities in alternative splicing have been considered as a novel shared mechanism implicated in the pathogenesis of mental illnesses [29]. As an illustration, an aberrant relative abundance of alternatively spliced RNA isoforms of the *SNCA*, *SRRM2*, *DAO*, *SHANK3*, *CACNA1C*, and *TSC2* genes has been observed in the brains of patients suffering from amyotrophic lateral sclerosis, autism spectrum disorder, and Parkinson’s disease [27,30,31]. Although recent studies in mice showed that gut bacteria could influence brain gene splicing [18,32], our comprehensive understanding of the molecular mechanisms by which gut bacteria-induced alternative splicing occurs remains lacking.

In this study, using published RNA-sequencing data from ten brain regions of rats orally treated with the probiotic Lacidofil^®^ [33], we systematically characterized probiotic-driven alternative splicing events and investigated their biological implications via GO/KEGG enrichment analysis, the protein–protein interaction network, psychiatric disorder association analysis, and comparison with gene expression alterations. Firstly, brain alternative splicing patterns were extensively modulated in the process of probiotic–host interaction, which could be attributed to the finely regulated splicing and expression of splicing factors. Secondly, the DSGs across distinct brain regions were consistently enriched in specific cellular complexes and pathways. Additionally, those in the LConly, mPOA, and venDG were significantly associated with psychiatric disorders. Thirdly, we illustrated the precise splicing of neurodegenerative-pathology-associated exons and the independent effect of probiotic supplementation on gene splicing and expression. Taken together, the findings of our study advance the understanding of the regulation of the host alternative splicing program by gut bacteria in the context of the central nervous system, which offers vital implications that potentially aid in the development of effective therapies to treat mis-splicing neuropathology.

## 2. Method

### 2.1. Description of Datasets Used

To investigate the regulatory effects of probiotics on host RNA splicing, we analyzed the RNA-Seq data of Long–Evans male rats obtained from Charles River (Charles River, Canada) in a recently published study. In the study, the experimental rats were treated with the probiotic Lacidofil^®^ (Lallemand, QC, Canada) in their drinking water. The daily dose was 1.5 × 10^9^ CFU per animal, and the concentration of probiotic solution was 10 g/L. The control rats were maintained on the purified water. After 21 days of treatment, brain tissue samples (*n* = 100) were collected from ten distinct regions: the basolateral amygdala (BLA); cingulate cortex (CGC); dorsal dentate gyrus (dorDG); infralimbic cortex (ILC); LConly; mPOA; nucleus accumbens shell (NACShell); prelimbic cortex (PLC); Raphe; and venDG. The RNA was extracted from these samples and sequenced on the Illumina NovaSeq 4000 platform with 151 bp reads. The sequencing data are available in the Gene Expression Omnibus (GEO) database under the accession number GSE222756 [33].

### 2.2. Gene Alternative Splicing and Expression Analysis

Quality control of the downloaded raw sequence data was performed using Fastp software (v0.20.1) [34], with a threshold of Q30 for the low-quality bases. Sequence alignment was conducted against the rat reference genome (*Rattus norvegicus*, Rnor_6.0.101) using Hisat2 software (v2.2.1) [35], followed by transcript assembly using StringTie software (v2.1.2) [36]. Differential alternative splicing events were identified using rMATs software (v4.1.0) [37], with events considered significant at *p* < 0.05. Differential gene expression analysis was performed using the DESeq2 software package (v1.26.0) [38], using the criteria of fold change ≥ 1.5 and *p* < 0.05.

### 2.3. Functional Enrichment Analysis

A functional enrichment analysis of DSGs was conducted using clusterProfiler (v3.14.3) [39], utilizing both the GO (https://geneontology.org/) (accessed on 31 July 2024) and KEGG (https://www.genome.jp/kegg/pathway.html) (accessed on 31 July 2024) databases. GO term enrichment analysis encompassed three categories: biological processes, molecular functions, and cellular components. Statistical significance for both GO terms and KEGG pathways was established at *p* < 0.05.

The protein–protein interaction networks of DSGs were constructed using the Metascape platform (http://metascape.org) (accessed on 21 August 2024) [40], integrating multiple interaction databases including InWeb_IM, BioGrid, STRING, and OmniPath. Highly interconnected subnetworks were subsequently identified using the Molecular Complex Detection (MCODE) algorithm. Each subnetwork was functionally annotated through GO enrichment analysis, with the most statistically significant pathway (lowest *p*-value) designated as the primary biological function of the corresponding subnetwork.

### 2.4. Psychiatric Disorder Association Analysis

The PsyGeNET (Psychiatric Disorders and Gene Association Network) is a specialized expert-curated knowledge repository that facilitates the exploration of molecular mechanisms underlying psychiatric disorders [41]. This database encompasses gene sets associated with multiple psychiatric diagnostic categories, including the following: depressive disorders; bipolar and related disorders; schizophrenia spectrum and other psychotic disorders; and substance use disorders (specifically, alcohol, cocaine, and cannabis). Additionally, it includes substance-induced psychiatric conditions, namely, substance-induced depressive disorder and substance-induced psychosis. The PsyGeNET database currently contains approximately 4000 validated gene–disorder associations across various psychiatric disorders.

Based on gene annotation information from the Ensembl (http://www.ensembl.org/) (accessed on 6 October 2024) and NCBI (https://www.ncbi.nlm.nih.gov/) (accessed on 6 October 2024) databases, rat gene symbols were converted to their human orthologs. Subsequently, an enrichment analysis of psychiatric disorder-associated genes was performed using the psygenet2r software package (1.36.0) [42], with statistical significance set at *p* < 0.05.

## 3. Results

### 3.1. Probiotics Promote Extensive Alternative Splicing Changes Across the Brain

To investigate the probiotic-mediated modulation of gene splicing across diverse brain regions, we analyzed deep RNA-seq data from ten brain regions isolated from rats fed with plain drinking water or Lacidofil^®^, a probiotic formulation composed of *Lactobacillus helveticus* R0052 and *Lactobacillus rhamnosus* R0011 [33,43]. Using a significance threshold of *p* < 0.05 for alternative splicing events between control and treatment samples, our study identified a total of 2941 differential alternative splicing events involving 1048 genes across all brain tissues examined (Appendix A). These alternative splicing events were systematically categorized into five types including alternative 3′ splice site (A3SS), alternative 5′ splice site (A5SS), MXE, retained intron (RI), and SE. Intriguingly, while the absolute number of differential splicing events varied among brain regions, with BLA, Raphe, and, particularly, dorDG exhibiting a higher frequency (Figure 1A), the proportional distribution of each event type remained almost consistent across all ten examined regions. Specifically, SE events emerged as the most prevalent, closely followed by MXE events, which accounted for approximately 20% of the total differential splicing events. In contrast, RI and A3SS events were observed less frequently. A5SS events proved to be the rarest, located at the lowest tier of the frequency spectrum (Figure 1B). These data indicate that probiotics may primarily regulate brain alternative splicing through SE and MXE events.

Considering the profound effects of the probiotic Lacidofil^®^ on gene splicing, we investigated the similarities and distinctions in DSGs among different brain regions. The results revealed that among the 1048 genes, a substantial proportion (55%, 576 genes) exhibited differential splicing in more than one brain region (Appendix A), which implies concerted splicing modulation in multiple brain parts. The systematic examination of common DSGs elucidated their involvement in distinct functional gene families: Dnaj; Eif; Ppp; SRSF; Ube2; Usp; Hnrnp; and Atp (Figure 2A). Of note, the Hnrnp family emerged as the most represented gene family, encompassing the largest number of gene members. Specifically, this family comprised nine genes: *Hnrnpa1*, *Hnrnpa2b1*, *Hnrnpab*, *Hnrnpc*, *Hnrnpd*, *Hnrnpdl*, *Hnrnph3*, *Hnrnpk*, and *Hnrnpr*. On the other hand, the Atp family followed as the second largest, including *Atp13a2*, *Atp2b1*, *Atp5mk*, *Atp5pd*, *Atp6v0d1*, *Atp6v1g2*, and *Atp6v1h* (Figure 2A). Importantly, these gene families are associated with neuroscience research. For example, studies have demonstrated that *Hnrnpa1* dysfunction contributes to neurodegeneration in multiple sclerosis experimental models [44], and mutations in *ATP13A2* have been established as the causative factor for Kufor-Rakeb syndrome, a rare form of juvenile-onset Parkinsonism [45]. There were 31 genes that exhibited extensive splicing alterations (~7/10 regions; Figure 2B; Appendix A), highlighting their potential significance in probiotic-induced neural modulation. Remarkably, the *Eif4a2* gene, known for its RNA helicase properties and role in mediating mRNA binding to ribosomes [46,47], showed splicing changes across all examined brain regions. Moreover, *Mobp* (Myelin-Associated Oligodendrocyte Basic Protein) demonstrated differential splicing in all areas except the LConly. Moreover, *Mbp* (Myelin Basic Protein) as well as *Qk* (QKI, KH domain containing RNA binding) exhibited altered splicing across eight brain regions (Figure 2B).

Next, we also observed 472 genes that exhibited region-specific splicing alterations (Appendix A). Notably, the BLA, dorDG, and Raphe demonstrated a higher prevalence of unique splicing events (Appendix A), indicating that these three areas were highly specifically responsive to the probiotic. Given that the regulation of alternative splicing frequently exhibits tissue-specific characteristics, the unique splicing pattern in an individual region is supposed to be intrinsically linked to the specialized functional roles of its respective neuroanatomical domains. Overall, we elucidate the commonalities and heterogeneity of splicing responses across brain regions, while identifying the key genes and even key gene families that mediate probiotic-induced splicing regulation in the brain.

### 3.2. Probiotics Modulate Brain Neural Signaling Pathways via Alternative Splicing

To elucidate the biological significance of probiotic-induced DSGs, we performed functional enrichment analysis using the GO and KEGG pathway databases. DSGs across brain regions were enriched in a diverse range of GO terms and KEGG pathways (Appendix A). GO analysis revealed seven predominant functional categories shared among most regions: genetic information transmission, protein localization and transport, enzyme activity, intracellular signal transduction, cell adhesion and cytoskeleton, cell fate, and nervous system-specific function (Figure 3). In parallel, KEGG pathway analysis demonstrated strong enrichment in three categories, encompassing neurodegenerative diseases, synaptic function and plasticity, and substance addiction (Figure 4). There were also unique molecular responses for individual brain regions (Appendix A). GABA-associated GO terms showed region-specific enrichment patterns, with the Raphe being exclusively enriched in gamma-aminobutyric acid secretion, while the venDG was enriched in GABA receptor complex and GABA receptor activity. Furthermore, the BLA exhibited a distinct immunological landscape, being the only region enriched in GO terms linked to hypersensitivity responses (types I, IIa, and III). With regard to KEGG pathways, we found that the PLC displayed the highest enrichment in hormone- and secretion-related pathways compared with other brain regions (Appendix A). These results therefore suggest remarkably complex interaction between the probiotic Lacidofil^®^ and the brain alternative splicing machinery relevant to crucial neural signaling pathways.

Protein interactions form the basis of cellular complexes and biological processes [48]. Here, we constructed protein–protein interaction networks using the DSGs of individual brain regions to investigate the functional units required for brain cells to respond to the probiotic Lacidofil^®^. Our analysis detected thirteen significantly interconnected subnetworks (MCODE modules) within the original interaction network of the dorDG, while three to seven modules were included in other brain regions (Figure 5; Appendix A). Despite such regional differences in the complexity of probiotic-mediated molecular interactions, the functional modules of distinct brain regions seemed to yield common molecular machines. It is particularly noteworthy that all brain regions were consistently enriched in RNA splicing-related modules comprising multiple spliceosomal components (Figure 5; Appendix A). For instance, the corresponding module in the PLC contained coding genes (*Snrpn* and *Snrpa1*) of spliceosomal small nuclear ribonucleoproteins, and genes encoding splicing factors, including *Hnrnpc*, *Hnrnpdl*, *Prpf19*, and *Srrm2*. Likewise, genes encoding another small nuclear ribonucleoprotein (*Snrnp35*), the splicing regulator (*Eif4a3*), and four additional splicing factors (*Hnrnpd*, *Hnrnpk*, *Srsf2*, and *Pcbp2*) were identified in the matching module of the Raphe (Figure 5). It is particularly notable that the genes encoding distinct types of spliceosomal components across brain regions showed similar distribution patterns, with splicing factor genes being the most numerous and diverse category (Appendix A). In total, these splicing factor genes included the hnRNP gene family, SR gene family, *Bud31*, *Pcbp2*, *Srrm2*, *Tra2b*, *Yju2*, and *Prpf19*. Among them, *Pcbp2* demonstrated the highest frequency, being distributed across eight brain regions (Appendix A). Furthermore, different members of the hnRNP gene family (such as *Hnrnpd* and *Hnrnpk*) and the SR gene family (such as *Srsf7* and *Srsf10*) frequently co-occurred in specific regions. In addition, most brain areas simultaneously harbored members belonging to the above two gene families (Figure 5; Appendix A).

We also identified genes encoding various components of protein complexes responsible for vesicle transport and mechanical force generation during this process within the modules of nearly all brain regions (Figure 5; Appendix A). In the mPOA and venDG, the clathrin coat assembly modules contained genes encoding the core components (*Clta*, *Cltb*) and adaptor proteins/auxiliary factors (*Picalm*, *Ap1b1*, *Ap3s1*, *Gak*) of the clathrin complex. Furthermore, three genes encoding TRAPP complex subunits (*Trappc1*, *Trappc4*, and *Trappc6b*) were identified within the COPII-mediated vesicle transport module of the dorDG. In addition, the synaptic vesicle lumen acidification module of this region contained four coding genes (*Atp6v1h*, *Atp6v1c1*, *Atp6v1g2*, and *Atp6v0d1*) of V-ATPase complex subunits (Figure 5). Importantly, the expression of *Atp6v1g2* is known to be significantly restricted to neurons [49], further highlighting the specific role of the V-ATPase complex in the nervous system [50]. These findings suggest that the probiotic Lacidofil^®^ may profoundly influence host brain cellular spliceosome complexes and vesicle cycling through the regulation of alternative splicing.

### 3.3. Probiotic DSGs Are Associated with Psychiatric Disorders and Their Risk Genes

Next, to evaluate whether DSGs across the examined brain regions were associated with psychiatric disorders, we performed an enrichment analysis of mental illness-associated genes recorded in the PsyGeNET database. The results demonstrated both similarities and dissimilarities in the distributions of DSGs associated with PsyGeNET disorders across brain regions. Compared with other disorders, schizophrenia presented the most gene–disorder associations across all brain regions. In contrast, the second most common gene–disorder association was exhibited in depressive disorders (Figure 6A). Statistically, DSG sets from the LConly and mPOA significantly overlapped with depression-associated genes (*p* = 0.043), while those from the VenDG were enriched in schizophrenia (*p* = 0.036; Figure 6B). These findings suggest that the splicing abnormalities in the VenDG region may play a crucial role in the pathogenesis of schizophrenia, whereas similar aberrations in the LConly and mPOA regions appear to be particularly relevant to the development of depression.

It has been reported that the splicing of the *APP*, *APOE*, *SNCA*, *MAPT*, *DAO*, *SHANK3*, and *CACNA1C* genes, which are recorded in the PsyGeNET database, were abnormal in patients with neurological diseases such as Parkinson’s disease, Alzheimer’s disease, and amyotrophic lateral sclerosis [27,30,51,52,53]. Surprisingly, our study revealed that probiotic administration significantly affected the splicing of the *App*, *Apoe*, and *Mapt* genes in various brain regions of rats (Figure 7). The *App* gene, which encodes a transmembrane protein influencing synaptogenesis and neuroplasticity [54,55], is particularly noteworthy. In rats, the full-length *App* isoform comprises 18 exons [56], with exon 7 encoding a Kunitz-type proteinase inhibitor implicated in Alzheimer’s brain pathology [57]. Here, exon 7 and exon 8 (OX-2 antigen domain-encoding region) [57] were spliced in a mutually exclusive manner in the CGC. This area tended to increase exon 7 inclusion and exon 8 skipping in response to the probiotic Lacidofil^®^. Apart from this splicing alteration, the LConly of experimental rats, compared to the controls, might have independently elevated the inclusion of exon 8. These data indicate that the probiotic exerts targeted regulatory effects on both the Kunitz-type proteinase inhibitor domain and the OX-2 antigen domain of the *App* gene. The *Apoe* gene, encoding a member of the apolipoprotein family capable of binding to the App protein, is another documented risk factor in Alzheimer’s disease pathologies [58]. Previously, intron 3 retention in the *Apoe* gene was proved to be a post-transcriptional regulation mechanism of the mRNA level in mouse neurons under excitotoxic injury [59]. Comparably, we found that exposure to the probiotic Lacidofil^®^ might potentially upregulate the retention of intron 2 in the *Apoe* gene, especially in the rat ILC. *Mapt*, which encodes tau proteins expressed throughout the nervous system [60], also showed complicated splicing regulation. In the Raphe, exons 3 and 5 were spliced in a mutually exclusive manner. After probiotic administration, this brain region was prone to increasing exon 3 inclusion and exon 5 skipping. In addition, compared with the controls, the dorDG of treatment rats elevated the inclusion of exon 4 (Figure 7).

In addition to the genes described above, *Srrm2*, a known splicing factor gene, also exhibited aberrant splicing in both the brain and blood samples of patients with Parkinson’s disease [61]. Here, we show that in the probiotic-treated rat, the alternative splicing of *Srrm2* changed in nearly all brain areas (Appendix A). For instance, this gene contained an exon with A3SS in the NACShell, meaning that the probiotic promoted the inclusion of the longer exon 14, while the shorter exon 14 had a higher inclusion level in the normal condition (Figure 7). Altogether, our findings highlight the potential of probiotics to modulate specific functional domains of brain proteins. Moreover, these results indicate that gut microbial dysbiosis may partly contribute to the aberrant splicing of genes crucial for neurological health in patients with brain disorders.

### 3.4. Limited Overlap Between Probiotic-Induced Splicing Events and Differential Gene Expression

A substantial body of evidence indicates that the coupling of gene expression and alternative splicing is a critical gene regulation step in various biological processes [62,63]. Here, we conducted a differential gene expression analysis and calculated the percentage of genes with significantly altered expression levels among those exhibiting differential alternative splicing in each brain region. Notably, the vast majority of alternative splicing events identified in the BLA, CGC, LConly, mPOA, NACShell, PLC, Raphe, and venDG were observed in the absence of global changes in gene expression. However, interestingly, the percentage values of the dorDG and ILC were as high as 60% and 62% (Table 1). Notable examples of genes include an ATP-dependent RNA helicase *Eif4a2* [47] and a precursor protein of the hippocampal cholinergic neurostimulatory peptide *Pebp1* [64], both of which exhibited dual regulation across six brain regions (Appendix A).

The differential expression of splicing factors plays a critical role in modulating splicing outcomes across diverse tissues and cellular conditions [65,66]. Our study detected the occurrence of splicing factors that simultaneously exhibited altered alternative splicing patterns and differential expression levels in all brain tissues. Some splicing factors showed consistent change patterns across multiple brain areas. For instance, *Srrm2* expression increased in the BLA, NACShell, and venDG, while *Rsrp1* expression decreased in the CGC, BLA, Raphe, and venDG (Appendix A). These consistent changes may represent a common response mechanism of the nervous system to the probiotic. Interestingly, however, we also identified certain splicing factors that exhibited opposite trends among different brain regions. A notable example was the set of five splicing factors (*Prpf40a*, *Srsf10*, *U2af1l4*, *Hnrnpab*, and *Hnrnpa2b1*) shared by the dorDG and the ILC, which showed contrasting expression changes in these two areas (Figure 8). These findings indicate that the autoregulatory feedback mechanisms of splicing factors seem to be a pivotal process in probiotic-induced gut–brain axis signaling and that probiotics might induce regionally ubiquitous and regionally specific splicing patterns via the precisely modulated expression of splicing factors across distinct brain regions.

## 4. Discussion

To date, research on the effects of host symbionts on brain gene splicing has been limited, with almost all studies focusing on changes in the epigenetic landscape, chromatin restructuring, microRNA profiles, and gene expression levels [21,25,26,67,68,69]. In this study, we investigated alternative splicing events using RNA-seq data from ten distinct brain regions of rats that were treated with a probiotic targeting the gut–brain axis [33]. Our results showed a wide range of probiotic-associated alternative splicing alterations across the brain. These alternative splicing events were enriched in genes that encode the components of vesicular transport complexes and the spliceosome, especially splicing factors. Importantly, many splicing factors simultaneously underwent changes in gene expression levels. Moreover, we discovered for the first time that probiotic supplementation can precisely regulate the alternative splicing patterns of genes, and even specific exons, that have been confirmed to be associated with mental disorders. This study sheds light on the role of gut probiotics in modulating the mammalian brain alternative splicing landscape.

In all brain regions examined, SE was the most prevalent alternative splicing event, comprising approximately 60% of all detected alternative splicing events. This result is in line with expectations based on eukaryotic molecular evolution. While the frequencies of individual alternative splicing types diverge across eukaryotic lineages, animals, in contrast to other eukaryotes, have evolved transcriptomes with exceptionally high frequencies of exon skipping due to the enhanced prevalence of SE-associated features within their genomes [70]. While the contribution of host symbionts to the coevolution process of SE incidence and the host genome remains to be determined, SE events seem to be the principal molecular mechanism by which gut probiotics expand host brain proteome and phenotypic complexity. In contrast to SE events, MXE events are generally rare in most biological contexts [71]. However, previous studies have demonstrated that these events occur with a remarkably high frequency in the mammalian brain [72]. For example, alcohol exposure could substantially disturb alternative splicing profiles in the central nucleus of the amygdala, superior frontal cortex, basolateral amygdala, and nucleus accumbens, with the majority of these splicing events attributed to changes in aberrant SE and MXE events [73]. In our work, MXE events represented the second most frequent alternative splicing event. This finding suggests that beyond their particular effects in the brain, MXE events may serve as a critical mediator in the gut–brain axis.

While it is known that dysregulation in gene expression often underlies the pathogenesis of psychiatric symptoms in mental illness [74], less is known about the splicing response pattern in the context of the central nervous system. In various rat brain regions, probiotic administration induced significant splicing alterations in genes associated with various psychiatric disorders existing in the PsyGeNET database, suggesting that the probiotic-induced DSGs were not randomly distributed in the gene regulatory network. Reassuringly, the probiotic even seemed to selectively determine the usage of alternative exons. A prominent example is the differential splicing of the *App* gene, which is a common risk factor for neurodegenerative disorders characterized by β-amyloid aggregation [75]. In neurons, exon 7 and exon 8 of the *App* gene, which encode the Kunitz-type proteinase inhibitor and OX-2 antigen domain, respectively, are splicing-susceptible exons [57]. There is evidence suggesting that the splicing defects of these two exons trigger elevated β-amyloid formation [76]. In the current study, not all exons of the rat *App* gene responded equally to the probiotic. Coincidentally, it was exactly exon 7 and exon 8 that demonstrated differential splicing in several brain regions when we compared the results in experimental rats to those in the control group. Thus, gut probiotics demonstrate therapeutic potential in precisely modulating the alternative splicing of neurological disease-associated exons in the mammalian brain.

The current work revealed that genes exhibiting differential splicing in response to probiotics infrequently showed simultaneous expression alterations across the majority of rat brain regions. A similar finding was also reported by a previous fecal microbiota transplantation study. Mice receiving gut microbiota from patients with autism, compared to those transplanted with microbiota from healthy donors, showed few changes in brain gene expression but presented abundant alternative splicing events [18]. Theoretically, the regulatory networks of gene expression and splicing were involved in independent evolutionary trajectories. As expected, splicing and expression tend to modulate distinct biological pathways during a species’ adaptation to environmental factors [77,78]. Thus, the overlap between differentially spliced and expressed genes is generally low [79]. Given the above, the observed difference between genes with splicing plasticity and those with expression plasticity in the rat brain during host–microbe interaction may be due to the complexity of host genetic architecture. Another interesting result is that we found that splicing factors were differentially spliced and expressed across all brain regions. These data are highly consistent with earlier research that showed that a splicing factor could modulate its own expression and cross-regulate the levels of many other splicing factors through positive or negative feedback loops triggered by binding to corresponding transcripts [80,81,82]. On the other hand, our results might also be a reflection of the properties of molecular regulatory networks: genes encoding spliceosomal or transcriptional machinery components demonstrate dual plasticity in splicing and expression [77]. Altogether, it seems that gut probiotics may maximize their functional diversity via the multilayered control of host gene regulatory networks.

Based on the plethora of DSGs across the brain, our data also indicate that distinct areas of the brain tended to respond divergently to gut probiotic administration because of their specialized functions. Here, the BLA exhibited a distinct immunological reaction, being the only region enriched in GO terms linked to hypersensitivity responses (types I, IIa, and III). Immune cells are regarded as key regulators of brain functions. The BLA is obligatory in regulating fear stimuli [83]. Recently, the activation of astrocytes within this brain region was found to positively correlate with fear response intensity [84]. In schizophrenia, abnormalities in the structure and function of the hippocampus have consistently been reported [85]. Compared to other hippocampal subregions, such as the subiculum and hippocampal tail, the dentate gyrus plays a fundamental role in the pathophysiology of schizophrenia [86]. In our study, only the DSG set in the venDG was significantly enriched in schizophrenia-associated genes recorded in the PsyGeNET database. Thus, we speculate that the region-specific alternative splicing patterns in the venDG may largely contribute to the dentate gyrus’ unique role in schizophrenia. The abundance of splicing events and their correlation with gene expression in the dorDG markedly differed from those in other brain regions. These findings suggest that the sensitivity and complexity of the probiotic-responsive splicing networks exhibit region-specific variations across the brain.

We compared our findings with the original study by Rayan et al., which investigated the effects of the probiotic Lacidofil^®^ on gene expression across different brain regions in rats [33]. Their research demonstrated that probiotics could modulate mRNA levels throughout the brain, with particularly strong effects in the hippocampus. Similarly, we found that the dorDG, a hippocampal subregion, exhibited the highest abundance of alternative splicing events among all examined regions. Moreover, in the mPOA and LConly regions, both the DEGs identified in the original study and the DSGs detected in our analysis were significantly associated with the same psychiatric disorder [33]. Our study not only confirms the reliability of the prior findings but also provides complementary insights by demonstrating the probiotic-mediated regulation of gene splicing.

Altogether, we demonstrate a new paradigm of transcriptional responses to gut probiotic action in the mammalian brain. However, our study has several important limitations to consider. The identified differential splicing events require experimental validation. Moreover, we need further research to investigate whether these splicing changes could affect protein structures and functions. Finally, a comprehensive exploration of how diverse gut bacteria influence gene splicing in the host brain would enhance our understanding of splicing regulation under the gut–brain axis.

## Figures and Tables

**Figure 1 microorganisms-13-00165-f001:**
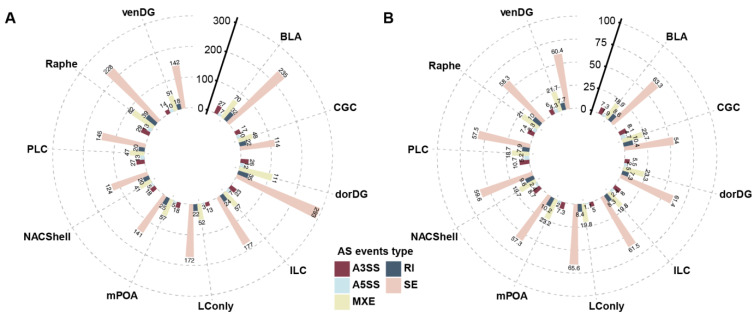
Probiotic-induced alternative splicing changes across the rat brain regions. (**A**,**B**) The number (**A**) and percentage (**B**) of the five types of significantly (*p* < 0.05) different alternative splicing events (bar color) are shown. AS is the abbreviation of alternative splicing.

**Figure 2 microorganisms-13-00165-f002:**
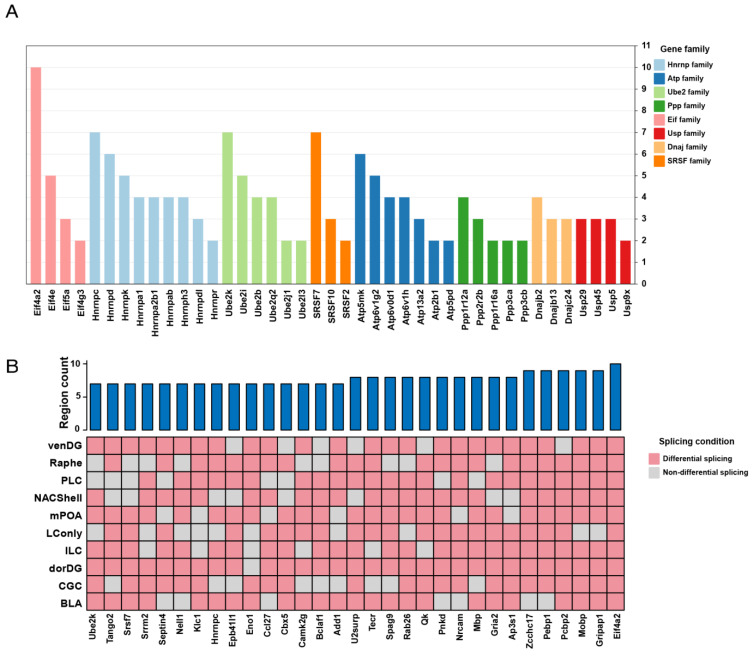
Shared DSGs across brain regions. (**A**) Gene family distribution of DSGs shared by at least two brain regions. The bar colors represent the types of gene families, and the bar heights represent the number of shared brain regions. (**B**) Distribution of genes showing widespread splicing pattern changes across brain regions. The heights of the blue bars represent the number of shared brain regions.

**Figure 3 microorganisms-13-00165-f003:**
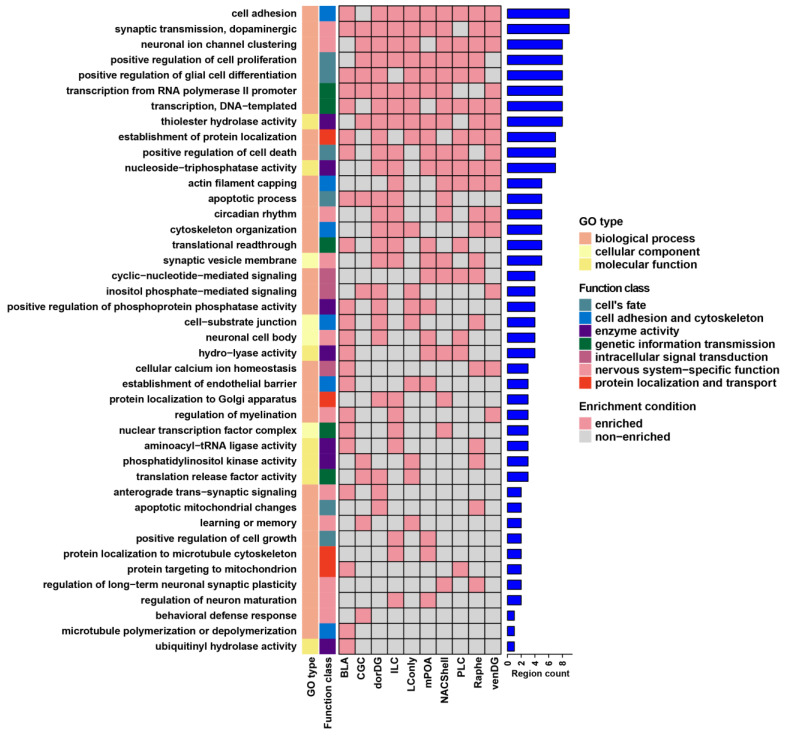
The distribution and classification of GO terms enriched in DSGs across brain regions. The colors in the leftmost bars denote different GO types and their function classes. The heights of the rightmost bars represent the number of shared brain regions.

**Figure 4 microorganisms-13-00165-f004:**
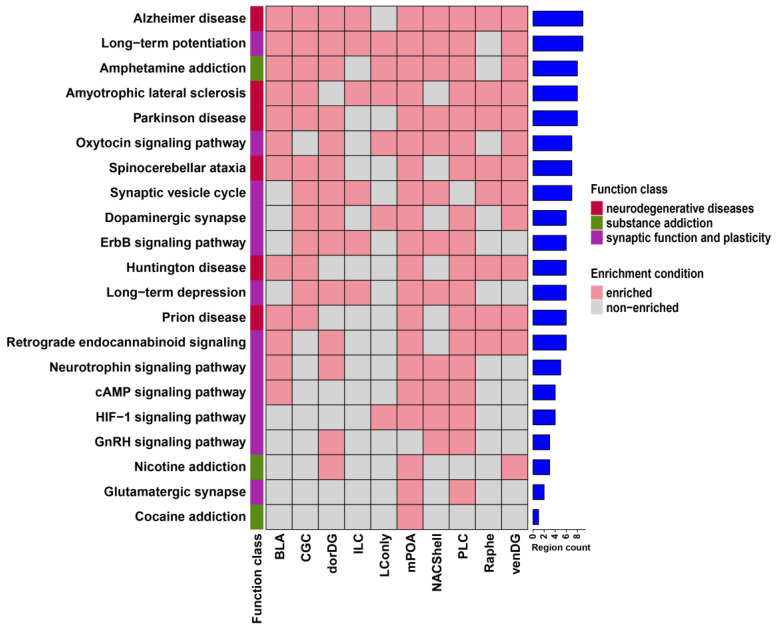
The distribution and classification of KEGG pathways enriched in DSGs across brain regions. The colors in the leftmost bars denote different function classes of KEGG pathways. The heights of the rightmost bars represent the number of shared brain regions.

**Figure 5 microorganisms-13-00165-f005:**
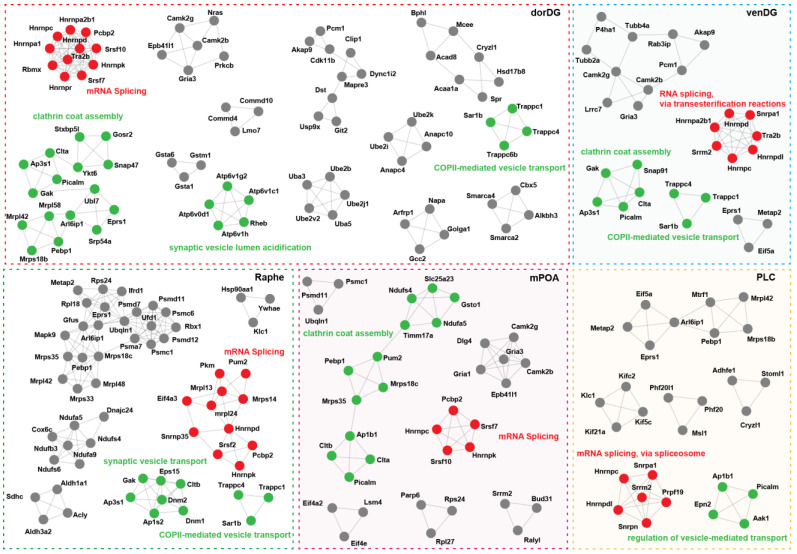
Visualization of densely connected networks for DSGs in dorDG, mPOA, PLC, Raphe and venDG. The MCODE modules highlighted with red nodes are involved in RNA splicing, while those with green nodes are associated with vesicle transport. The red or green text annotations denote the biological functions of the adjacent networks.

**Figure 6 microorganisms-13-00165-f006:**
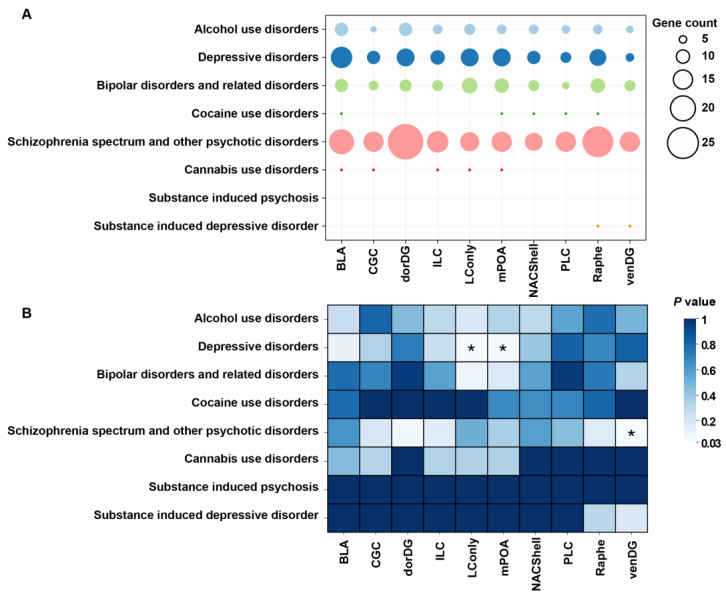
The validated associations between DSGs and psychiatric disorders. (**A**) The colors of the bubbles indicate the types of psychiatric disorders, while the sizes of the bubbles represent the number of associated genes. (**B**) The significance of associations between DSGs and psychiatric disorders. The colors of the squares represent the *p*-value. * *p*-value < 0.05.

**Figure 7 microorganisms-13-00165-f007:**
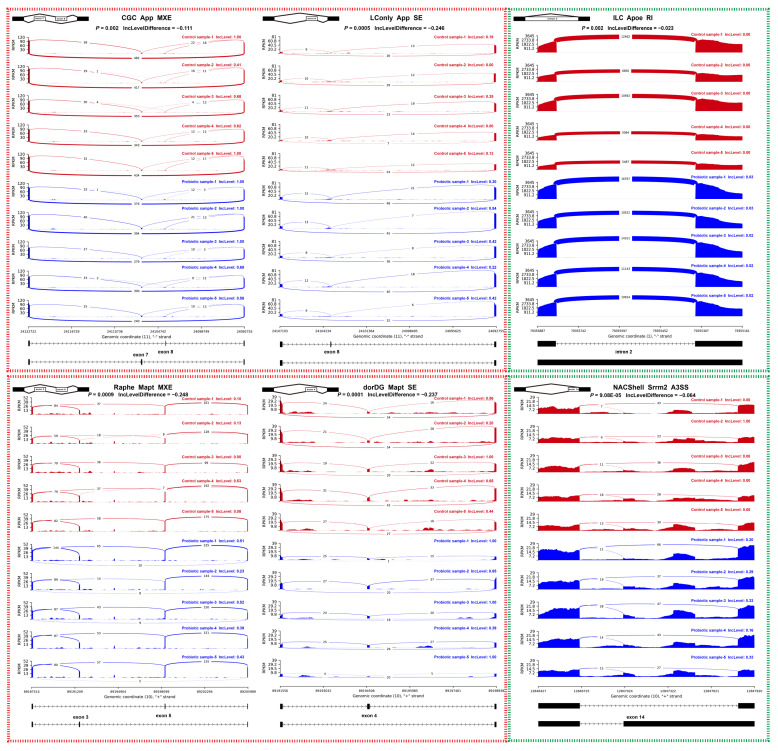
Visualization of splice junctions in alternative splicing events. Ten sashimi plots denote biological replicates for the control group (red) and probiotic group (blue). Numbers on the curved lines indicate the count of junction-spanning reads. The inclusion level (IncLevel) of each biological replicate, the differential average inclusion level between the control and probiotic groups (IncLevelDifference), and the corresponding *p*-value are demonstrated in the figure. The schematic diagrams at the top indicate the types of RNA splicing events. The black tracks at the bottom indicate the genomic locations of splicing events.

**Figure 8 microorganisms-13-00165-f008:**
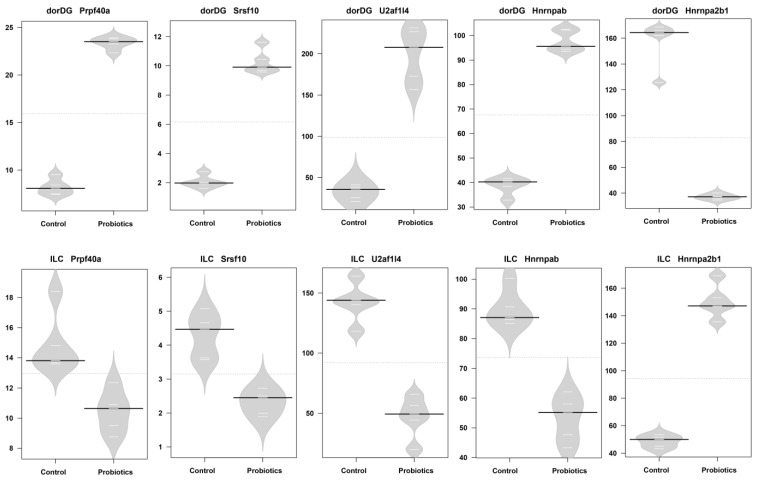
Expression profiles of selected splicing factors in the dorDG and ILC regions. Bean plots show the differential expression levels of splicing factor-encoding genes (*Prpf40a*, *Srsf10*, *U2af1l4*, *Hnrnpab*, and *Hnrnpa2b1*) between the control and probiotic groups. Black lines indicate the median values; white lines represent the data points; and polygons illustrate the estimated density of the data.

**Table 1 microorganisms-13-00165-t001:** Overlap between DSGs and DEGs across brain regions.

	BLA	CGC	dorDG	ILC	LConly	mPOA	NACShell	PLC	Raphe	venDG
**DSGs**	314	186	369	247	225	200	174	221	309	190
**DSGs vs. DEGs**	42	20	220	154	43	39	17	38	37	49
**Overlap Ratio**	13%	11%	60%	62%	19%	20%	10%	17%	12%	26%

DSGs—the number of DSGs; DSGs vs. DEGs—the number of genes in common between DSGs and DEGs; overlap ratio—the ratio of genes in common to DSGs.

## Data Availability

The public raw sequence data were retrieved from the GEO database under accession number GSE222756.

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
