# Peer review of "Changes in RNA Splicing: A New Paradigm of Transcriptional Responses to Probiotic Action in the Mammalian Brain"

_microorganisms, 2025, doi:10.3390/microorganisms13010165_

Round 1

Reviewer 1 Report

Comments and Suggestions for Authors

I thank the authors for interesting and well written manuscript. The manuscript offers interesting insights into the RNA splicing changes induced by probiotics in the rat brain, utilizing RNA sequencing data from a previously published study (Rayan et al., 2024). The research question is relevant and has the potential to contribute to the understanding of the gut-brain axis. However, several issues need to be addressed to improve the clarity and impact of the findings.

Specific Comments

The current presentation of the Figure 7 is not clear and lacks detailed labels and makes it difficult for interpretation. It is recommended that the figure would be revised to include clearer annotations and enhanced color contrasts to distinguish between different groups effectively.

The manuscript mentions the concentration of probiotics used and the general route of administration but does not specify the exact daily volume administered to the animals. Methods part should be very clear for the reader if it was by gavage or drinking water, not to be found this information only in the Results part. This detail is crucial for understanding the dosage and replicating the study. Providing the total CFU per animal per day would greatly enhance the reproducibility and clarity of the experimental design.

There are several instances of missing articles ("the" and "a") like the gut-brain axis.

Discussion Section

The discussion lacks a thorough comparison with the findings of the original study by Rayan et al. It is essential to highlight how the current findings align with or differ from the original study to contextualize the new data within the broader research landscape.

The manuscript does not discuss the limitations of the study, which is essential for a balanced scientific argument. Discussing potential biases, the limitations of analysis and obtained results, and the generalizability of the findings would provide a more complete understanding of the study's implications and reliability.

The manuscript would benefit from a more detailed integration of its findings with existing literature on gut-brain interactions and RNA splicing. Highlighting novel mechanisms or pathways identified and discussing their potential implications for understanding or treating neurological conditions would enhance the manuscript's contribution to the field.

Reviewer 2 Report

Comments and Suggestions for Authors

This work represents a fascinating approach to explaining the phenomena attributed to the gut-brain axis modification by probiotics.   The use of RNA seq data and the different bioinformatic approaches demonstrate the occurrence of alternative splicing, providing a mechanistic explanation for some pathological states.

The work and its analysis is sound and extensive.  However, 

-Some formal details should be addressed: scientific names (elegans  should be C. elegans,  Drosophila, should use capital letters).

- I think there is an abuse of the use of abbreviations,  so it is sometimes difficult to follow the ideas

- Numbers should substitute citations in the text to facilitate reading
